



# Plant trait response of tundra shrubs to permafrost thaw and nutrient addition

Maitane Iturrate-Garcia[1], Monique M.P.D. Heijmans[2], J. Hans C. Cornelissen[3], Fritz H. Schweingruber[4], Pascal A. Niklaus[1], Gabriela Schaepman-Strub[1]

[1]Department of Evolutionary Biology and Environmental Studies, University of Zurich, Zurich, 8057, Switzerland
[2]Plant Ecology and Nature Conservation, Wageningen University & Research, Wageningen, 6700 AA, the Netherlands
[3]Systems Ecology, Department of Ecological Sciences, Vrije Universiteit Amsterdam, Amsterdam, 1081 HV, the Netherlands
[4]Swiss Federal Research Institute WSL, Birmensdorf, 8903, Switzerland

*Correspondence to*: Maitane Iturrate-Garcia (maitane.iturrate@gmail.com)

**Abstract.** Plant traits reflect growth strategies and trade-offs in response to environmental conditions. Because of climate warming, plant traits might adapt, altering ecosystem functions and vegetation–climate interactions. Despite important feedbacks of plant trait changes in tundra ecosystems with regional climate, with a key role for shrubs, information on responses of shrub functional traits is limited.

Here, we investigate the effects of experimentally increased permafrost thaw depth and (possibly thaw-associated) soil nutrient availability on plant functional traits and strategies of arctic shrubs in northeastern Siberia. We hypothesize that shrubs will generally shift their strategy from efficient conservation to faster acquisition of resources through adaptation of leaf and stem traits in a coordinated whole-plant fashion. Thereto we ran a 4-year permafrost thaw and nutrient fertilization experiment with a fully factorial block design and six treatment combinations − permafrost thaw (control, unheated cable, heated cable) × fertilization (no-nutrient addition, nutrient addition). We measured ten leaf and stem traits related to growth, defence and the resource economics spectrum in four shrub species, which were sampled in the experimental plots. The plant trait data were statistically analysed using linear mixed-effect models and principal component analysis (PCA).

The response to increased permafrost thaw was not significant for most shrub traits. However, all shrubs responded to the fertilization treatment, despite decreased thaw depth and soil temperature in fertilized plots. Shrubs tended to grow taller, but did not increase their stem density or bark thickness. We found a similar coordinated trait response for all four species at leaf and plant level, i.e. they shifted from a conservative towards a more acquisitive resource economy strategy upon fertilization. In accordance, results point towards a lower investment into defence mechanisms, and hence increased shrub vulnerability to herbivory and climate extremes.

Compared to biomass and height only, detailed data involving individual plant organ traits such as leaf area and nutrient contents or stem water content can contribute to a better mechanistic understanding of feedbacks between shrub growth strategies, permafrost thaw and carbon and energy fluxes. In combination with observational data, these experimental tundra



trait data allow for a more realistic representation of tundra shrubs in dynamic vegetation models and robust prediction of ecosystem functions and related climate-vegetation-permafrost feedbacks.

## 1 Introduction

Plants have different strategies to use resources to grow, reproduce, compete with neighbour plants and defend themselves against pathogens and herbivores (Bazzaz et al. 1987; Ordoñez et al. 2010). However, as resources – nutrients, water, and light – can have limited availability, plants are subject to trade-offs so that they have to allocate the resources to one function versus another (Grime 1977; Westoby et al. 2002; Reich 2014). Environmental changes, such as the ones promoted by climate warming (e.g. increasing amount of resources available in the soil), may modify these trade-offs and plant strategies

(Grime 2006; Ordoñez et al. 2010).

Plant strategies and trade-offs can be identified by measuring plant traits and their correlations (Grime et al. 1997; Westoby et al. 2002). The so-called leaf economics spectrum is an example of how leaf traits show similar covariation across species according to the resource acquisition strategy, which goes from slow (i.e. resource conservation) to rapid resource acquisition. Plant traits also determine plant responses to environmental factors and underpin their effects on ecosystem

processes and services (Lavorel & Garnier 2002; Kattge et al. 2011; Soudzilovskaia et al. 2013). Analysing plant trait responses to climate warming can provide insight into future ecosystem structure and functioning (Díaz et al. 2007).

In low-resource environments such as the Arctic tundra, plants adopt a conservative strategy with low rates of resource acquisition, growth and tissue turnover (Chapin 1980). Low stature, small leaves of low specific leaf area (thick leaves with dense tissue) and long leaf life span reflect that conservative strategy (Reich, Walters & Ellsworth 1997; Cornelissen 1999).

This strategy allows plants to allocate resources to other processes, such as defence against pathogens and herbivores, and confer stress resistance (Chapin, Autumn & Pugnaire 1993). Plants however might adopt a more acquisitive strategy under the environmental conditions projected for the Arctic in the course of this century.

Surface air temperature in the Arctic has risen more rapidly than in other regions over the past decades and is projected to keep increasing (~3°C by the year 2100 under emission scenario RCP4.5) (IPCC 2013). Ground heat flux and soil

temperature are also expected to increase, accelerating permafrost thaw and soil organic matter mineralization (Rustad et al., 2001; Richter-Menge & Overland, 2010; Elmendorf et al., 2012). The release of nutrients trapped in the permafrost (Kokelj & Burn 2003; Weintraub & Schimel 2003; Schuur et al. 2009), together with an enhanced soil mineralization rate (Schmidt, Jonasson & Michelsen 1999; Walther et al. 2002), will increase nutrient availability for tundra vegetation (Keuper et al. 2012). Several experiments and satellite imagery have shown effects of recent climate warming on tundra vegetation growth,

productivity and distribution, especially on shrubs (Myers-Smith et al. 2011, 2015; Elmendorf et al. 2012; Tape et al. 2012; Myers-Smith & Hik 2018). Our current knowledge of tundra shrub responses to climate change concerns mainly their performance traits sensu Violle et al. (2007), especially plant height and biomass. However, we still know precious little about the functional traits underpinning these responses or the effects expanding shrubs may have on ecosystem functions



(but see Hudson, Henry & Cornwell 2011; Kremers, Hollister & Oberbauer 2015; Barrett & Hollister 2016). A recent pan-
arctic plant trait sampling and analysis effort has revealed a generally strong spatial temperature-trait relationship, which was
however mediated by soil moisture (Bjorkman et al., 2018a, Bjorkman et al. 2018b). This study also highlighted the
limitations of the observational space-for-time substitution method and identified the need for experimental studies to
elucidate intraspecific trait responses to environmental drivers. Shrub responses to climate may have consequences for the
carbon cycle (e.g. increase carbon uptake) and the surface energy budget (e.g. decrease albedo), which in turn may affect the
regional climate (Eugster et al., 2000; Chapin, 2003; Beringer et al., 2005; Bonfils et al., 2012; Pearson et al., 2013; Juszak et
al., 2017). A better understanding of shrub trait responses to climate, and shrub–climate interactions is fundamental to
improve dynamic global vegetation models and predictions of vegetation shifts (Cramer et al. 2001; Doherty et al. 2010;
Wullschleger et al. 2014).

In this study, we experimentally investigate the consequences of increased permafrost thaw and nutrient addition on
aboveground traits and trait coordination of tundra shrubs. We hypothesize that under simulated future environmental
conditions (i.e. permafrost thaw and soil nutrient increase), (i) shrubs will shift their strategy from efficient conservation to
faster acquisition of resources through adaptation of leaf and stem traits; and (ii) leaf traits, stem traits and height will show a
coordinated response to these environmental changes as they all belong to the same overall resource economy dimension
within the functional trait space. To test our hypotheses we ran a permafrost thaw and fertilization experiment for four years
in Siberia, and measured ten plant traits related to the leaf economics spectrum, growth and defence in tundra shrubs.
Whereas most previous studies focused on one or two shrub species only, we explicitly compare the responses of four
predominant species in order to find commonalities versus idiosyncrasies of intra- and interspecific trait response, as these
are critical for upscaling from site level to tundra ecosystems at larger scales.

## 2 Materials and methods

### 2.1 Study area

The study area is located in the nature reserve of Kytalyk, in the continuous permafrost region of Yakutia, northeastern
Siberia (70°49'N, 147°28'E, 10 m a.s.l.). Ice-rich permafrost and shallow active layers characterize the area (van Huissteden
*et al.,* 2005; Iwahana *et al.*, 2014). The mean annual precipitation is 210 mm and the mean annual air temperature -13.1°C,
with minimum and maximum monthly means of -33.5°C in January and 11.2°C in July (1980–2013, WMO station 21946,
Chokurdakh, monthly summaries of GHCN-D, NOAA National Climatic Data Center).

The experimental plots were placed on a moist acidic tussock tundra area, the soil of which is classified as Gelisol (Wang *et al.* 2017). In the Circumpolar Arctic Vegetation Map (Raynolds *et al.*, 2019) the vegetation type in this area is classified as
tussock-sedge, dwarf-shrub, moss tundra. The main vegetation has a maximum canopy height of 25 cm and comprises sedge
allies (mainly *Eriophorum vaginatum*), abundant deciduous and evergreen dwarf shrubs, bryophytes and lichens.   The



slightly acidic soil (pH 6) has a silty-clay texture and high organic matter content (Blok *et al.* 2010, Bartholomeus *et al.*, 2012). The soil organic matter decomposition is low as indicated by the high average carbon to nitrogen ratio (22) and low cellulose to lignin ratio (2.4) (Iturrate-Garcia *et al.* 2016). At mid-growing season, the mean active layer thickness is 35 cm, increasing to about 50 cm at the end of the season.

## 2.2 Experimental design

To test whether climate change might have effects on shrub traits, we ran a permafrost thaw and nutrient fertilization experiment from 2011 to 2014 (Wang *et al.* 2017). The experiment had a fully factorial block design with five blocks, each with six plots of $1.5 \times 1.5$ m placed at randomly chosen locations. Six treatment combinations – permafrost thaw (3 levels) $\times$ fertilization (2 levels) – were randomly assigned to the plots within blocks.

We buried heating cables powered by solar panels at approximately 15 cm depth to increase the thaw depth. The permafrost
thaw treatment consisted of no cable, unheated cable and heated cable. The unheated cable plots served as a reference for the permafrost thaw treatment, while plots without cable were included to quantify possible disturbance effects of the cable alone. For the fertilization treatment (nutrient addition versus no addition), we applied slow-release NPK fertilizer tablets with micronutrients (Osmocote Exact Tablet, Scotts International, Heerlen, the Netherlands). The tablets were applied at approximately 5 cm depth at the start of the experiment and two years later (5.6 g N, 1.4 g P and 3.7 g K $\cdot$ m$^{-2}\cdot$ yr$^{-1}$), which
increased the exchangeable nutrient content mainly in the upper soil layer (Wang et al., 2017).

## 2.3 Soil temperature and thaw depth

Soil temperature of each plot was measured in 2013-2014 at four depths (0 cm, 5 cm, 15 cm and 25 cm) using temperature loggers (iButton DS1922L/DS1921G, Maxim Integrated, USA). Thaw depth was measured twice in July 2014 by introducing vertically a metal rod with centimetre scale until hitting the frozen soil (Wang *et al.* 2017).

**2.4 Study species and sampling**

We investigated the response of four shrub species dominant at the study site and present in all experimental plots: two deciduous species *Betula nana* ssp. *exilis* (Sukazcev) Hultén and *Salix pulchra* Cham., and two evergreen species *Ledum palustre* ssp. *decumbens* (Aiton) Hultén and *Vaccinium vitis-idaea* L. (Fig. 1). The abundance of the four species was broadly similar in all plots, except for *S. pulchra*, which was less abundant.
We randomly selected six healthy-looking individuals (with less than 20% leaf damage) of each species per plot at mid-growing season, except for *S. pulchra* for which only one to four individuals were present per plot. We cut the selected individuals 4 cm below the root collar after measuring their height. The sampling and transport of the plant samples followed the protocol for standardised trait measurements described in Pérez-Harguindeguy *et al.* (2013). Most of the plant traits were measured in the laboratory within a few hours.



### 2.5 Plant traits

We selected ten aboveground plant traits, which are related to and provide insight into shrub growth, defence and nutrient acquisition strategies, as well as into the interactions between tundra shrubs and carbon and energy fluxes. We measured the selected leaf and stem traits in each individual of the four shrub species one single time (i.e. growing season of the last year of experiment).

### 2.5.1 Height

Plant height was measured in the field as the vertical distance from the ground to the tallest vegetative tissue of the selected individuals (maximum vegetative height).

### 2.5.2 Leaf area (LA) and specific leaf area (SLA)

We cut two leaves per individual, including the petiole, from the top and bottom canopy layers. We scanned the leaves with a flatbed scanner (LiDE 70 Canon Inc., Japan, 300 dpi image resolution) calibrated with a $1cm^2$ reference. Then, we estimated LA by counting pixels using the software MatLab R2014a (The MathWorks, Inc., MA, USA). We oven-dried the scanned leaves (60°C, 72 h) and weighed them to determine SLA by dividing the LA of each leaf by its dry weight.

### 2.5.3 Leaf dry matter content (LDMC)

We followed a variation of the partial rehydration method to determine LDMC using the same leaves as for LA (Vendramini *et al.* 2002; Vaieretti *et al.* 2007). To assure maximum hydration, we cut whole individuals in the morning, wrapped the samples in moist paper and put them in sealed plastic bags (Pérez-Harguindeguy *et al.* 2013). We kept the samples in the dark at low temperatures until they were weighed within the following six hours to obtain fresh mass. The individual leaves were re-weighed after oven-drying them (60°C, 72 h). LDMC was the dry mass of a leaf divided by its fresh mass.

### 2.5.4 Leaf nitrogen concentration (LNC)

Oven-dried leaves were milled and leaf carbon and nitrogen concentrations determined by dry combustion (TruSpec Micro-CHN analyser, Leco Corporation, MI, USA) in samples of 2 mg. Then, the carbon to nitrogen ratio (C:N) was calculated.

### 2.5.5 Leaf phosphorus concentration (LPC)

We used a colorimetric assay employing ammonium heptamolybdate to determine LPC. Milled samples of 0.05 g were combusted in a muffle furnace (B180 Nabertherm, Germany) programmed with one-hour heating up ramp to 600°C and two hours and a half at 600°C. We added 2 ml of 0.1 M $H_2SO_4$ to the ashes, followed by 5 ml of distilled water, and filtered the suspension (Macherey Nagel MN615). The phosphorus in the extracts was determined using a continuous flow analyser (Skalar Analytical B.V., the Netherlands) calibrated with $KH_2PO_4$ standards.





### 2.5.6 Stem-specific density (SSD)

We cut approximately 3 cm long sections of the main stem at one third of the stem length and removed the bark. We
measured the diameter and length of the stem sections without bark, oven-dried (60°C, 72 h) and weighed them. SSD was
determined by dividing the dry mass of a section by its volume.

### 2.5.7 Stem water content (SWC)

We weighed the sections used for SSD before and after oven-drying them. SWC was estimated as the difference between
fresh and dry weight divided by the dry weight.

### 2.5.8 Xylem diameter and bark thickness

Samples including the 2 cm above and below the root collar of the main stem were taken and preserved in ethanol (40% vol.
aqueous solution) until laboratory processing. We cut thin sections of 20 – 30 μm along the root collar of each individual and
placed them on microscope slides. We photographed and measured xylem diameter and bark thickness following the
protocol described in Iturrate-Garcia *et al.* (2017).

### 2.6 Statistical analysis

To test if soil temperature and thaw depth were affected by permafrost thaw and fertilization treatments, we used linear
mixed-effect models fitted in asreml (ASReml 3.0, VSN International Ltd., UK). The fixed terms of the models were block
(factor with five levels) and the interaction among permafrost thaw treatment (two levels: heating, no-heating) and
fertilization treatment (two levels). For the analysis, we averaged the thaw depth values per plot. The soil temperature values
were averaged by growing and no-growing season per plot and depth class.
We also used linear mixed-effect models to test the treatment effect on plant traits. Height, LA, bark thickness, and xylem
diameter were log-transformed prior the statistical analysis. First, we analysed plant traits of the four species together and
then plant trait of each functional type (PFT; deciduous and evergreen). The fixed terms of the models were block, the
interaction among permafrost thaw treatment (three levels), fertilization treatment (two levels) and species (four levels; first
analysis) or PFT (two levels) and species (PFT analysis). For the first analysis, we also considered the interaction between
species and block (term recognised in the course of the statistical analysis to take into account species-specific trait
differences among blocks). The random terms were plot (factor with 30 levels) and the interaction of plot and species. In
both cases, we tested for effects of increased thaw depth and cable disturbance by splitting the three-level permafrost thaw
factor into two contrasts of one degree of freedom (df) each, i.e. presence of cable and heating. Cable effects were tested by
fitting heating (heated cable vs. unheated cable and no cable) followed by cable (heated and unheated cable vs no cable),
whereas increased thaw depth effects were tested by fitting cable presence followed by heating. We found that plant traits
were significantly different among species, even between species within the same PFT. Consequently, we analysed the four





species separately to maintain ecological information. We fitted block and the interaction between the permafrost thaw and fertilization treatments as fixed terms and plot as a random term.

In order to explore shrub plant strategy and its change with treatments, standardized (Z-scored) plant trait data were subjected to a principal component analysis (PCA; vegan package version 2.4-0; Oksanen 2016). We only considered the fertilization treatment (nutrient addition and no addition) in the PCA, as most traits were not responsive to the permafrost thaw treatment (see Results). We performed a separate analysis for leaf traits (SLA, LDMC, LNC, LPC and C:N), and one for stem traits (SSD, bark thickness, xylem diameter and SWC) and height. Scores and variable loadings resulting from the

PCA were scaled for data visual depiction.

To test for relationships between leaf economics, and stem traits and height, we used linear mixed-models. We extracted the loadings of the first principal component axes (PC1) of the leaf trait and the stem trait-height PCA. The response variables in our models were height and stem trait-height PC1 loadings. Block and leaf trait PC1 loadings were set as fixed terms and plot as random effect. The significance of the linear relationships between variables was analysed using Pearson's

correlation coefficients in addition to the linear mixed-effect models.

All data were analysed using R 3.4.1 (http://r-project.org).

## 3 Results

### 3.1 Treatment effects on soil temperature and thaw depth

Permafrost thaw and fertilization treatments affected soil temperature and thaw depth (Table 1). The soil temperature during

the growing season was significantly higher in heated plots than in unheated plots. The soil temperature at 5 cm depth was 0.6°C higher ($F_{1,29} = 9.00$, p < 0.05), at 15 cm 1.1°C ($F_{1,29} = 17.97$, p < 0.001) and at 25 cm 0.8°C ($F_{1,29} = 14.02$, p < 0.01). The difference of surface soil temperature (0 cm) between heated and unheated plots was not significant. During the no-growing season, temperature differences were significant at all the depths. The soil temperature at 0 cm was 1.4°C higher in heated plots than in unheated plots ($F_{1,29} = 16.2$, p < 0.01), at 5 cm 1.3°C ($F_{1,29} = 26.1$, p < 0.001), at 15 cm 1.3°C ($F_{1,29} = $

17.5, p < 0.01) and at 25 cm 1.2°C ($F_{1,29} = 16.9$, p < 0.01). The difference in soil temperature between fertilized and unfertilized plots was also significant at all the depths, but only during the growing season. The soil temperature was lower in the fertilized plots: at 0 cm 0.9°C lower ($F_{1,29} = 11.6$, p < 0.01), at 5 cm 1.0°C ($F_{1,29} = 19.1$, p < 0.01), at 15 cm 0.4°C ($F_{1,29} = 12.7$, p < 0.01) and at 25 cm 0.5°C ($F_{1,29} = 6.24$, p < 0.05). The thaw depth was 10.7 cm deeper in heated plots than in unheated plots ($F_{1,29} = 24.6$, p < 0.001), but 3.9 cm shallower in fertilized plots than in unfertilized plots ($F_{1,29} = 5.40$, p <

0.05). Fertilization treatment effects on soil temperature and thaw depth did not depend on the permafrost thaw treatment.

### 3.2 Treatment effects on leaf and stem traits and plant height

The permafrost thaw treatment had no significant effect on most shrub traits. Only LA responded significantly to the permafrost thaw treatment ($F_{1,28} = 18$, p < 0.001) when analyzing all four species together. At the species level, the





permafrost thaw treatment affected only LA of *S. pulchra* and *L. palustre*. Individuals of both species had greater LA on

heated plots than on control and unheated plots (Table 2). The permafrost thaw treatment only increased SWC for *S. pulchra* ($F_{1,28} = 12.8$, p < 0.01). Neither the effect of the combination of treatments (permafrost thaw × fertilization) nor the disturbance caused by the buried cables were significant for most measured leaf and stem traits. Exceptions were a significant treatment combination effect on bark thickness of *B. nana* ($F_{1,25} = 4.54$, p < 0.05) and *L. palustre* ($F_{1,25} = 8.15$, p < 0.01), and LA of *S. pulchra* being negatively affected by the buried cables (Table 2).

The fertilization treatment had a significant effect on all leaf traits, height, and SWC, but not on bark thickness, xylem diameter or SSD, when the four species were analysed together (results not shown). At the PFT level, traits were significantly different between deciduous and evergreen species, except for LDMC and SSD. We also found that the fertilization effects on LA, LNC, LPC, C:N and SWC differed between PFTs (Table S2). The relative increase of LA and decrease of C:N with fertilization was greater for evergreen than for deciduous species. For LNC, LPC, and SWC, the

increase was greater for deciduous than for evergreen species. At the species level, the fertilization effect on LA, LNC and C:N was significant for all four shrub species (Table 2). Fertilization effects were also significant for SLA, LPC and LDMC of all species except for *S. pulchra* (Table 2). Leaves in the fertilized plots were bigger and thinner (higher SLA), had higher nutrient concentration (LNC, LPC) and lower LDMC and C:N than leaves in unfertilized plots (Table 2). For stem traits, the fertilization treatment significantly increased the SSD of *B. nana* ($F_{1,29} = 10.1$, p < 0.01) and SWC of both deciduous species

(*B. nana*: $F_{1,29} = 17.8$, p < 0.001; *S. pulchra*: $F_{1,29} = 13.9$, p < 0.01). Xylem diameter and bark thickness responses to nutrient addition were not significant.

### 3.3 Coordinated trait response to fertilization

In the leaf trait PCA with all four species combined, shrub individuals were separated into species with low overlap along the first principal component axis (PC1) (Fig. 2). PC1 explained 64% of the variation among individuals and was mainly

related to leaf nutrient content (LNC, LPC) and C:N. We found *B. nana* at the lower end of PC1, associated with high SLA and leaf nutrient concentrations, and *V. vitis-idaea* at the upper end of the axis. *B. nana* was the species with the widest range along PC1. The second PC axis (PC2) explained 19% of the variation and was mainly related to LDMC. Under nutrient addition, we observed a similar trait change for all four species. Leaves on fertilized plots had lower LDMC and C:N and higher LNC, LPC and SLA than leaves on unfertilized plots (Fig. 2).

Similar leaf trait space occupation was found when we ran the PCA for each species separately (Fig. 3). PC1 explained a slightly greater amount of total variance among individuals for the evergreen species (*L. palustre* 65% and *V. vitis-idaea* 60%) than for the deciduous species (*B. nana* 54% and *S. pulchra* 41%). Individuals were separated into two clusters along PC1 corresponding to individuals from fertilized and unfertilized plots. PC2 explained 17% and 18% of the variation among individuals for *L. palustre* and *V. vitis-idaea*, respectively, and 20% for deciduous species. The main results were maintained

when we excluded LNC from above analysis, showing that PC1 was not driven by the potential correlation of C:N and LNC (Table S3).





Similarly, we ran a PCA for stem traits and plant height for each of the four species (Fig. 4). For these traits, individuals overlapped more on the PCA ordination plane. However, there was a trend towards taller individuals having lower SSD and higher SWC in the fertilized plots for three species, but not for *V. vitis-idaea*. Indeed the stem trait-height space was generally similar for all the species except *V. vitis-idaea*. PC1 explained slightly more variation among individuals than PC2, especially for deciduous species.

### 3.4 Plant strategies – correlation of leaf traits with stem traits and plant height

We found significant correlation between PC1 of the leaf trait PCA (leaf PC1) and plant height for all species, except for *S. pulchra* (Fig. 5). We also found a significant correlation between leaf PC1 and stem trait-height PC1 for *B. nana* and *V. vitis-idaea* (Fig. S1). Individuals found in the upper range of the stem trait-height PC1 (high values for height, xylem diameter and bark thickness) were also found on the upper extreme of leaf PC1 (high values of LNC and LPC).

### 4 Discussion

We experimentally tested the effects of increased thaw depth and nutrient availability on plant traits of four tundra shrub species. While no strong responses to permafrost thaw were observed, our findings did show a coordinated response of leaf traits to fertilization, i.e. from a strategy of conservation of resources towards more rapid resource acquisition at leaf level, as we had hypothesised. Stem traits also tended towards a coordinated response to fertilization, though to a lesser extent. Moreover, one of the two deciduous (i.e., *Betula nana*) and one of the two evergreen species (i.e., *Vaccinium vitis-idaea*) showed a coordinated response of leaf and stem traits to fertilization along the same resource economics axis.

### 4.1 Treatment effects on plant traits

We expected that permafrost thaw and fertilization treatments would affect plant traits. However, our results showed that most of the plant traits responded only to the shallow nutrient addition. Plant growth in high-latitude ecosystems is highly nutrient-limited and so is shrub growth (Billings & Mooney 1968; Shaver & Chapin 1980; Epstein *et al.* 2000). Nutrient addition releases shrubs from this limitation and promotes their growth and biomass production (Chapin & Shaver 1996; DeMarco *et al.* 2014; Iturrate-Garcia *et al.* 2017). The shrub growth limitation release by adding nutrients is reflected in the plant trait changes we found, such as greater height, SLA and leaf nutrient concentration (Hudson, Henry & Cornwell 2011; Reich 2014). It is notable that in this short-term, strong fertilization treatment reduced soil temperatures and permafrost depth were measured. Despite less favourable soil physical conditions, shrubs followed a more acquisitive growth strategy under fertilization as compared to the permafrost thaw treatment which had higher soil temperatures and deeper thaw depth. The fact that plant traits were less responsive to permafrost thaw than to fertilization might be explained by the relatively large amount of nutrients added to the plots with the fertilization treatment. This amount may be greater than the nutrient release by permafrost thaw (Giblin *et al.* 1991; Hartley *et al.* 1999; Schaeffer *et al.* 2013). Furthermore, the depth of soil



layers at which nutrients were available for plants and of shrub rooting might also explain the different trait responses to the treatments. Most of the root biomass of the shrub species studied occurs at shallow soil depth (ca. 5-10 cm), which is shallower than the permafrost thaw depth during the growing season (Churchland *et al.* 2010; Keuper *et al.* 2012; Wang *et al.* 2017). In addition to permafrost thaw and related release of nutrients, soil warming might also enhance the nutrient availability through acceleration of soil organic matter mineralization (Knorr *et al.* 2005). Hartley et al. (1999) found effects of soil warming on subarctic shrub growth by using heating cables buried at 5 cm depth, which increased the soil temperature by 5°C. In our study, however, the heating cables were buried at 15 cm below the surface. Thereby, most warming was in the mineral soil layers below 15 cm, whereas the increase of soil temperature in the shrub root layer was lower than the threshold (1°C or greater) needed for increasing nutrient mineralization (Schmidt *et al.* 1999).

### 4.2 Coordinated leaf trait response to nutrient addition

Resource availability is thought to be one of the main drivers of plant strategy selection (Grime 2006; Ordoñez *et al.* 2010). In arctic tundra, where resource availability is low, shrub species adopt a conservative strategy with slow growth and tissue turnover, which enhances plant survival under harsh conditions (Chapin *et al.* 1993). However, the "slow traits" associated with the conservative strategy are disadvantageous in case of higher resource availability as shrub species could be outcompeted (e.g. through shading) by other species with faster growth and biomass production (Reich 2014). Our results show that species with similar resource economic strategies cluster into groups – deciduous and evergreen plant functional types – defined by their covarying leaf traits (Reich *et al.* 1997, 1999). On unfertilized plots, the deciduous shrub species *B. nana* and *S. pulchra* were characterized by leaf traits associated with faster resource acquisition: high SLA and leaf nutrient concentration and low LDMC and C:N. In contrast, the evergreen shrub species *L. palustre* and *V. vitis-idaea* were characterized by leaf traits associated with resource conservation, as expected due to a slower tissue turnover as compared to deciduous shrubs (Chapin & Shaver 1996).

We found different plant trait responses to fertilization with PFT for most leaf traits. Despite these differences, the increase of nutrients promoted a common coordinated response of leaf traits of all species, which reflects a change in resource economics from conservation to faster acquisition, even in the case of the evergreen species. Thus, there appears to be a comparable shift towards resource acquisitiveness in the leaf economics spectrum both between species, i.e. from evergreen to deciduous (Wright *et al.* 2004; Freschet *et al.* 2010; Díaz *et al.* 2016)*,* and within species (this study, Aerts *et al.* 2012). Since deciduous shrubs have been found to expand much more than evergreen shrubs in biomass and abundance in response to fertilization, both in Eurasian and North American tundra (van Wijk *et al.* 2003), our findings point to a possible important positive feedback between species turnover and intraspecific change with respect to resource economics traits.

### 4.3 Stem traits response to nutrient addition

Stem traits were less responsive to treatments than leaf traits, which might be explained by the relative short term of the experiment. Turnover of wood tissue is slower than of leaf tissue (Negrón-Juárez *et al.* 2015). Thus, stem traits might require



more time to show responses. Furthermore, the age heterogeneity of the selected shrubs might mask stem trait responses.

Older individuals have higher SSD and bark thickness than younger ones (Woodcock & Shier 2002; Patiño *et al.* 2009; Poorter *et al.* 2014). Therefore, stem trait responses might become statistically significant when longer-term experiments are run and shrubs within the same age class (i.e. similar stem diameter) are selected.

Under nutrient addition, we found that coordinated stem trait response tended towards greater height and SWC and lower SSD. These findings are in line with previous studies showing a negative relationship between wood density and water

content (Dias & Marenco 2014). Stems with lower SSD have less space filled with cell walls than those with higher SSD and therefore more water can be stored within the stem wood (McCulloh *et al.* 2011; Dias & Marenco 2014). Woody species with denser wood grow slower, have less wood water content and produce smaller and thicker leaves, which might be associated with a whole-plant strategy (Bucci *et al.* 2004; Wright *et al.* 2004; Ishida *et al.* 2008; Chave *et al.* 2009). However, our results showed that coordination between stem-height PC1 and leaf PC1, which reflect different trade-offs,

were only significant for half of the species. At stem and leaf levels, functional trade-offs may operate partly independently (Fortunel, Fine & Baraloto 2012).

### 4.4 To grow or to defend

Our findings suggest that shrubs will grow taller, acquire more resources and allocate them to produce larger leaves at lower cost (thinner leaves with lower LDMC and C:N). These changes in plant traits, together with an expected faster growth, will

come with a cost for shrubs: a decrease of their stress resistance (growth–defence trade-off) (Chapin *et al.* 1993; Chave *et al.* 2009; Iturrate-Garcia *et al.* 2017). The faster resource acquisition will make shrubs more vulnerable to herbivory due to higher leaf nitrogen content (Mattson 1980; Díaz *et al.* 2016) and to adverse environmental conditions (i.e. low nutrient availability) as consequence of low nutrient tissue reserves (Reich 2014). We also found that shrubs with more rapid resource acquisition grew taller but without increasing their bark thickness and SSD, which might enhance shrub

vulnerability to pests, mechanical and hydraulic failure, and extreme climatic events (Baraloto *et al.* 2010; Reich 2014; Díaz *et al.* 2016).

### 4.5 Shrub−climate feedbacks

Vegetation is strongly coupled with environmental conditions (Wookey et al. 2009; Medinski et al. 2010). Shrubs will be affected by climate warming, with resultant changes in plant strategy and traits, affecting species diversity and ecosystem

functions, such as carbon cycling and the surface radiation budget (Chapin *et al.* 1996; Beringer *et al.* 2005; Myers-Smith *et al.* 2011). The carbon uptake associated with increasing shrub growth and biomass production together with longer turn-over time due to carbon storage in branches as compared to leaf material will affect the carbon cycle (Hobbie, Nadelhoffer & Högberg 2002; Mack *et al.* 2004). Moreover, shrub trait changes may as well affect ecosystem processes. The production of low-cost tissues might accelerate litter decomposition because these tissues are easier to decompose than expensive ones

(McLaren et al. 2017).





Our results suggest that tundra shrubs will be affected by increased nutrient availability in shallow soil layers. Deeper-rooting species, such as graminoids, may benefit more from nutrient release by permafrost thaw in deep soil layers (Keuper *et al.* 2017; Wang *et al.* 2017). In competition with graminoids, shrubs will pre-empt nutrient and light resources by growing faster and taller, producing denser canopies and leaves with greater photosynthetic area (Chapin & Shaver 1996; Hudson *et al.* 2011; Elmendorf *et al.* 2012; Díaz *et al.* 2016). Bryophyte and lichen diversity is expected to decline due to the increase of shading and litter deposition associated with those changes (Cornelissen *et al.* 2001; van Wijk *et al.* 2003; Elmendorf *et al.* 2012; Lang *et al.* 2012). As a consequence of the cryptogam decrease, the thermal insulation of the permafrost might be reduced (Blok *et al.*, 2011a), promoting permafrost thaw and the release of carbon (e.g. in the form of methane) to the atmosphere (Schuur *et al.* 2008; Schaefer *et al.* 2011). However, also shrub cover increase has been reported to reduce summer permafrost thaw locally (Blok *et al.*, 2011b; Nauta *et al.*, 2015; Wang *et al.*, 2017). While these studies discussed shading effects as main cause, our detailed trait analysis suggests additional mechanisms associated with water demand. Shrubs under nutrient addition showed greater SLA, lower LDMC and higher water content of leaves and stems pointing towards enhanced water demand through higher photosynthetic potential and evapotranspiration. The higher water demand might deplete soil water resources, as suggested by the lower soil moisture and summer soil temperature in the fertilized plots (for detailed results see supplementary material in Wang *et al.*, 2017), where deciduous shrubs increased most. This depletion might result in reduced soil moisture, heat conductivity, heat flux and temperature and in turn reduced permafrost thaw. Shading by increased shrub cover might therefore not be the only driver of the reduced permafrost thaw. Water demand by plants, especially shrubs, might be at least as important, as also documented in Juszak *et al.*, 2016. Interestingly, soil moisture has been found as a potential growth co-limiting factor of tundra shrubs (Blok *et al.*, 2010, Myers-Smith *et al.*, 2015). However, shrubs might be released from water limitation by the predicted concomitant increase in precipitation. Thereby, related effects on shrub growth, community composition and feedbacks with the permafrost system and the atmosphere remain to be tested.

## 5 Conclusions

The climatic conditions projected for the Arctic, the shrub growth sensitivity to climate, and the importance of shrub–climate feedbacks for ecosystem functioning suggest that a special effort should be made to better understand future tundra changes and adaptation to the new climatic conditions. Here, we presented the response of a wide set of traits of selected dominant species in tussock tundra to permafrost thaw and increased nutrient availability. This response can be considered a step towards more realistic dynamic global vegetation models, although generalization should be considered cautiously due to the short term of the response, the spatial heterogeneity of Arctic regions and the complexity of shrub–climate feedbacks. According to our results, coordinated trait responses representing the whole plant (including wood and bark traits as in our study and ideally also root traits) instead of single trait responses are needed for a more robust prediction of shifts in vegetation, ecosystem processes and related climate–vegetation feedbacks.



**Data availability**

All data presented in this paper will be available in the DryAd repository.

**Author contributions**

MI and GS conceived the idea and methods of the study; GS and PAN obtained the grant that funded this research. MMPDH conceived the experimental design; MI collected the data; FHS instructed and contributed to the dendroecological work; MI and PAN analysed the data; MI led the writing of the manuscript; GS, PAN, MMPDH and JHCC contributed critically to the drafts. All authors gave final approval for publication.

**Competing interests**

The authors declare that they have no conflict of interest.

**Acknowledgements**

We acknowledge T.C. Maximov and his team from the Institute for Biological Problems of the Cryolithozone, Siberian Branch of the Russian Academy of Science, for the logistical support and Kytalyk Nature Reserve for permission to conduct
our research. We also thank I. Juszak for developing the MatLab code to calculate leaf areas, R. Simeon for preparing the samples for the CHN analysis, P. Wang for sharing the soil temperature and thaw depth data, J. Oehri for her help analysing the abiotic data, and J. Kattge for helpful comments on an earlier draft.

**Financial support**

This study was supported by the University Research Priority Program on Global Change and Biodiversity of the University
of Zurich (URPP-GCB), the Swiss National Foundation (SNSF project grant 140631) and the Netherlands Organisation for Scientific Research (NWO-ALW, VIDI grant 864.09.014).

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





**Table 1: Average soil temperature (standard deviation) and average thaw depth (standard deviation) of the five experimental blocks for plots grouped into no-fertilization (NF) and fertilization treatments (F). Data are grouped by growing season ($T_{Jun-Aug}$) and no-growing season ($T_{Sep-May}$) and depth (0, 5, 15 and 25 cm).**

|  | NH | H | NF | F |
|---|---|---|---|---|
| $T_{Jun-Aug}$ (°C) |  |  |  |  |
| 0 cm | 8.4 (1.9) | 8.5 (2.2) | 8.8 (2.1) | **7.9**[**] **(1.8)** |
| 5 cm | 4.7 (1.9) | **5.3**[*] **(2.0)** | 5.4 (2.1) | **4.4**[**] **(1.8)** |
| 15 cm | 1.4 (1.4) | **2.5**[***] **(1.7)** | 2.2 (1.6) | **1.8**[**] **(1.4)** |
| 25 cm | 0.4 (1.2) | **1.2**[**] **(1.5)** | 0.9 (1.5) | **0.4**[*] **(1.2)** |
| $T_{Sep.-May}$ (°C) |  |  |  |  |
| 0 cm | -13.1 (7.2) | **-11.7**[**] **(7.0)** | -12.4 (7.2) | -12.7 (7.1) |
| 5 cm | -11.8 (6.8) | **-10.5**[***] **(6.7)** | -11.2 (6.9) | -11.4 (6.8) |
| 15 cm | -11.0 (6.7) | **-9.7**[***] **(6.6)** | -10.5 (6.7) | -10.6 (6.7) |
| 25 cm | -10.6 (6.6) | **-9.4**[**] **(6.5)** | -10.1 (6.6) | -10.2 (6.6) |
| Thaw depth (cm) |  |  |  |  |
|  | 37.0 (6.6) | **47.7**[***] **(3.2)** | 42.5 (7.6) | **38.6**[*] **(7.2)** |

[*]$P < 0.05$   [**]$P < 0.01$   [***]$P < 0.001$





**Table 2: Effects of fertilization (Fert), cable disturbance (Ca) and soil heating (H) on leaf traits of each shrub species (*B. nana* (Betn), *S. pulchra* (Salp), *L. palustre* (Ledp) and *V. vitis-idaea* (Vacv)). Treatment columns show the average and standard error of the response variables for no fertilization (NFert), fertilization (Fert), no heating cables (Ct), unheated cables (Ca) and heated cables (H). LMM columns show the Wald test outputs for our linear-mixed models (*** P < 0.001, ** P < 0.01 and * P < 0.05). Significant effects are in bold. Treatment combination effect (heating × nutrient addition) was not included as it was significant**
**only for LA of Betn ($F_{1,25}$ = 15.1 , P < 0.01).**

| | Treatment | | | | | LMM | | |
|---|---|---|---|---|---|---|---|---|
| | Fertilization | | Permafrost thaw | | | Fert | Ca | H |
| | NFert | Fert | Ct | Ca | H | $F_{1,29}$ | $F_{1,28}$ | $F_{1,28}$ |
| **Leaf area (cm$^2$)** | | | | | | | | |
| Betn | $0.98 \pm 0.02$ | $1.08 \pm 0.02$ | $0.98 \pm 0.02$ | $1.04 \pm 0.03$ | $1.07 \pm 0.03$ | **7.96*** | 1.38 | 0.55 |
| Salp | $3.20 \pm 0.14$ | $4.26 \pm 0.24$ | $3.85 \pm 0.23$ | $3.38 \pm 0.19$ | $4.01 \pm 0.35$ | **22.7*** | **5.42*** | **6.68*** |
| Ledp | $0.27 \pm 0.01$ | $0.43 \pm 0.01$ | $0.35 \pm 0.02$ | $0.33 \pm 0.02$ | $0.37 \pm 0.02$ | **146*** | 1.02 | **6.73*** |
| Vacv | $0.39 \pm 0.01$ | $0.63 \pm 0.02$ | $0.48 \pm 0.03$ | $0.51 \pm 0.03$ | $0.54 \pm 0.02$ | **62.0*** | 0.85 | 0.51 |
| **Specific leaf area (cm$^2 \cdot$ g$^{-1}$)** | | | | | | | | |
| Betn | $133.3 \pm 2.1$ | $158.5 \pm 3.3$ | $144.7 \pm 3.7$ | $148.1 \pm 4.2$ | $144.9 \pm 3.2$ | **19.3*** | 0.24 | 0.22 |
| Salp | $122.5 \pm 2.5$ | $125.5 \pm 3.5$ | $122.2 \pm 3.8$ | $122.6 \pm 3.9$ | $127.3 \pm 3.7$ | 0.19 | 0.00 | 0.84 |
| Ledp | $54.6 \pm 1.1$ | $62.1 \pm 1.3$ | $56.2 \pm 1.5$ | $56.8 \pm 1.7$ | $62.1 \pm 1.2$ | **10.2**** | 0.04 | 3.42 |
| Vacv | $59.5 \pm 1.5$ | $80.4 \pm 1.9$ | $68.4 \pm 2.5$ | $69.0 \pm 2.3$ | $72.6 \pm 2.6$ | **71.7*** | 0.00 | 1.40 |
| **Leaf dry matter content (g $\cdot$ g$^{-1}$)** | | | | | | | | |
| Betn | $0.55 \pm 0.01$ | $0.45 \pm 0.01$ | $0.51 \pm 0.02$ | $0.49 \pm 0.02$ | $0.50 \pm 0.02$ | **14.8*** | 0.67 | 0.25 |
| Salp | $0.50 \pm 0.01$ | $0.44 \pm 0.02$ | $0.48 \pm 0.02$ | $0.47 \pm 0.02$ | $0.45 \pm 0.02$ | 2.00 | 0.01 | 0.36 |
| Ledp | $0.54 \pm 0.01$ | $0.48 \pm 0.01$ | $0.52 \pm 0.01$ | $0.51 \pm 0.01$ | $0.49 \pm 0.01$ | **38.6*** | 0.60 | 2.30 |
| Vacv | $0.53 \pm 0.01$ | $0.47 \pm 0.01$ | $0.51 \pm 0.01$ | $0.51 \pm 0.01$ | $0.49 \pm 0.01$ | **12.7**** | 0.10 | 1.60 |
| **Leaf nitrogen content (%)** | | | | | | | | |
| Betn | $24.2 \pm 0.5$ | $32.8 \pm 0.8$ | $28.3 \pm 1.0$ | $29.1 \pm 1.2$ | $28.0 \pm 1.2$ | **61.6*** | 0.33 | 0.63 |
| Salp | $16.6 \pm 0.6$ | $22.4 \pm 0.7$ | $20.6 \pm 0.7$ | $18.3 \pm 1.0$ | $19.9 \pm 1.1$ | **26.2*** | 2.58 | 1.63 |
| Ledp | $14.4 \pm 0.5$ | $18.2 \pm 0.4$ | $15.5 \pm 0.6$ | $16.4 \pm 0.6$ | $16.9 \pm 0.8$ | **27.3*** | 1.09 | 0.22 |
| Vacv | $7.8 \pm 0.2$ | $11.0 \pm 0.6$ | $8.6 \pm 0.4$ | $9.3 \pm 0.6$ | $10.2 \pm 0.8$ | **28.0*** | 1.03 | 1.47 |
| **Leaf phosphorus content (mg$^1 \cdot$ g$^{-1}$)** | | | | | | | | |
| Betn | $2.05 \pm 0.08$ | $3.90 \pm 0.19$ | $2.95 \pm 0.24$ | $2.94 \pm 0.24$ | $3.03 \pm 0.27$ | **60.6*** | 0.00 | 0.09 |
| Salp | $1.57 \pm 0.12$ | $1.55 \pm 0.01$ | $1.52 \pm 0.10$ | $1.32 \pm 0.09$ | $1.81 \pm 0.17$ | 0.00 | 0.17 | 3.92 |
| Ledp | $1.02 \pm 0.05$ | $1.32 \pm 0.05$ | $1.07 \pm 0.06$ | $1.15 \pm 0.05$ | $1.29 \pm 0.09$ | **13.9**** | 0.63 | 2.02 |
| Vacv | $0.59 \pm 0.03$ | $0.80 \pm 0.04$ | $0.64 \pm 0.04$ | $0.68 \pm 0.05$ | $0.75 \pm 0.05$ | **21.8*** | 0.39 | 1.63 |
| **Leaf carbon to nitrogen ratio** | | | | | | | | |
| Betn | $20.9 \pm 0.4$ | $15.8 \pm 0.6$ | $18.2 \pm 0.7$ | $18.1 \pm 0.9$ | $18.8 \pm 0.9$ | **28.5*** | 0.02 | 0.33 |
| Salp | $29.0 \pm 1.4$ | $22.6 \pm 0.8$ | $24.0 \pm 0.9$ | $27.7 \pm 1.8$ | $25.3 \pm 1.8$ | **10.9**** | 2.26 | 1.32 |
| Ledp | $38.0 \pm 0.9$ | $29.0 \pm 0.7$ | $35.3 \pm 1.4$ | $33.3 \pm 1.2$ | $32.1 \pm 1.2$ | **53.0*** | 1.91 | 0.57 |
| Vacv | $66.6 \pm 1.9$ | $49.7 \pm 2.0$ | $62.0 \pm 2.7$ | $58.8 \pm 3.0$ | $54.1 \pm 2.7$ | **45.6*** | 1.03 | 2.38 |








**Figure 1: Study species:** *Betula nana* **(a),** *Salix pulchra* **(b),** *Ledum palustre* **(c) and** *Vaccinium vitis-idaea* **(d).**









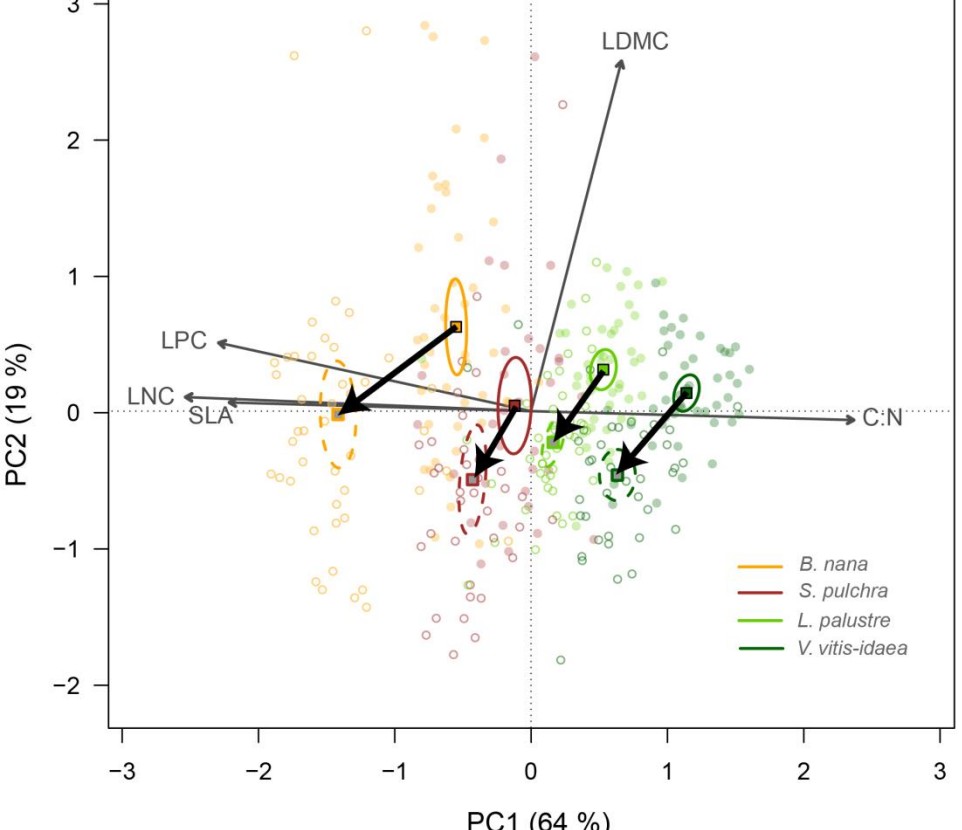



**Figure 2: Principal component biplot of leaf traits for all four shrub species combined, showing change along the leaf conservative-acquisitive continuum (black thick arrows) when nutrients were added. Leaf traits included are leaf dry matter content (LDMC), carbon to nitrogen ratio (C:N), leaf nitrogen content (LNC), leaf phosphorus content (LPC) and specific leaf area (SLA). Points are the trait scores of individuals without fertilization (closed circles) and with fertilization (open circles). Sample scores are scaled by factor 15 and variable loadings by factor 7. Squares indicate the centre of the ordiellipses (standard error with 95% confidence interval) of trait scores without nutrient addition (solid lines) and with nutrient addition (dashed lines). The first principal component explains 64% of the total variance, while the second component explains 19%.**








**Figure 3: Principal component biplots of leaf traits for each shrub species. Change in leaf traits when nutrients are added is shown by black thick arrows. Leaf traits included in the PCA are leaf dry matter content (LDMC), carbon to nitrogen ratio (C:N), leaf nitrogen content (LNC), leaf phosphorus content (LPC) and specific leaf area (SLA). Points are the trait scores of individuals without fertilization (closed circles) and with fertilization (open circles). Sample scores are scaled by factor 15 and variable loadings by factor 7. Squares indicate the centre of the ordiellipses (standard error with 95% confidence interval) of the trait scores without nutrient addition (solid lines) and with nutrient addition (dashed lines). The total variance explained by the two first principal components (PC1, PC2) is indicated as percentage between brackets on the axes.**


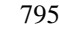



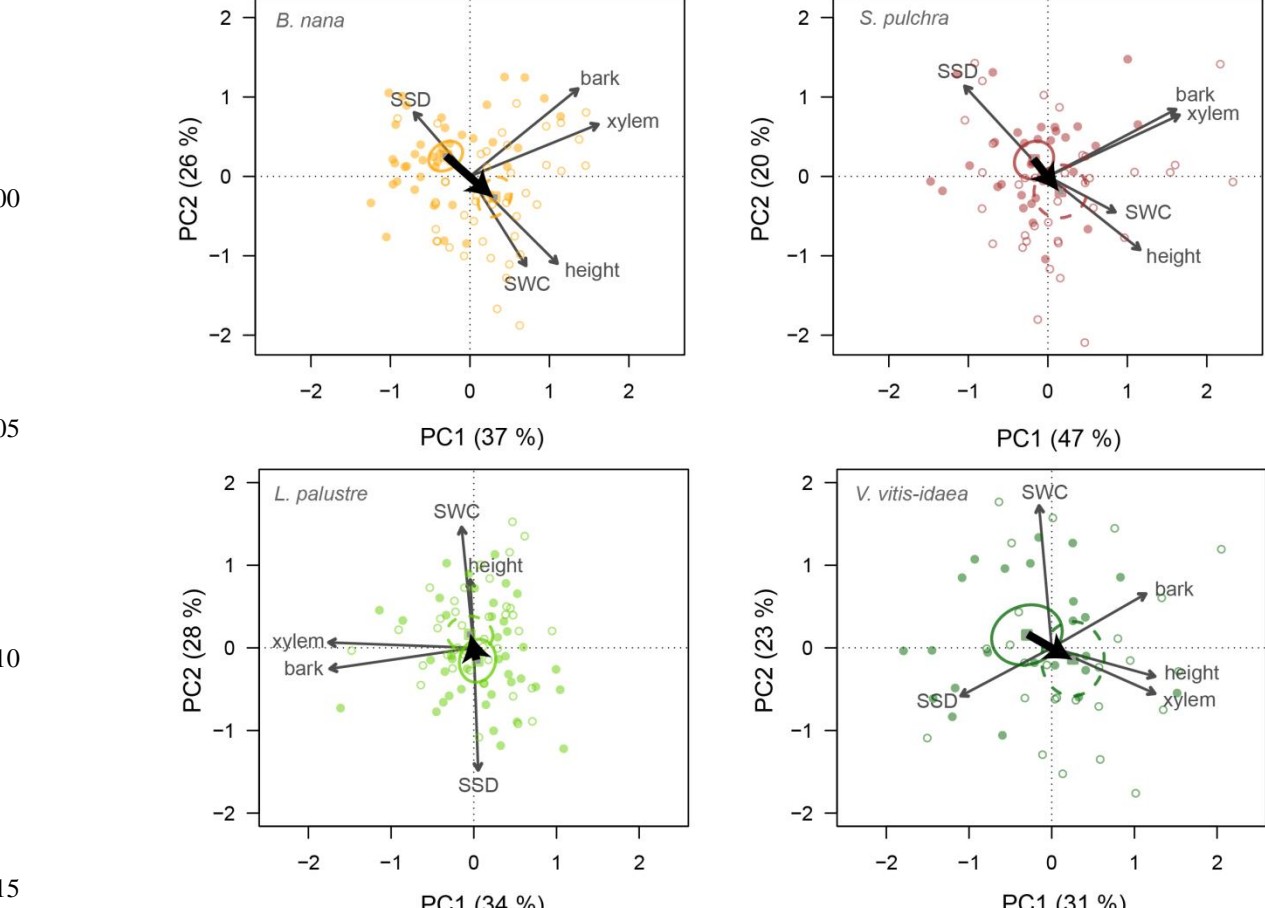





**Figure 4: Principal component biplots of plant height and stem traits for each shrub species. Stem traits included are xylem diameter, bark thickness, stem water content (SWC), and stem-specific density (SSD). Change in traits when nutrients are added is shown by black thick arrows. Points are the trait scores of individuals without fertilization (closed circles) and with fertilization (open circles). Sample scores are scaled by factor 15 and variable loadings by factor 7. Squares indicate the centre of the ordiellipses (standard error with 95% confidence interval) of the trait scores without nutrient addition (solid lines) and with nutrient addition (dashed lines). The total variance explained by the two first principal components (PC1, PC2) is indicated as percentage between brackets on the axes.**




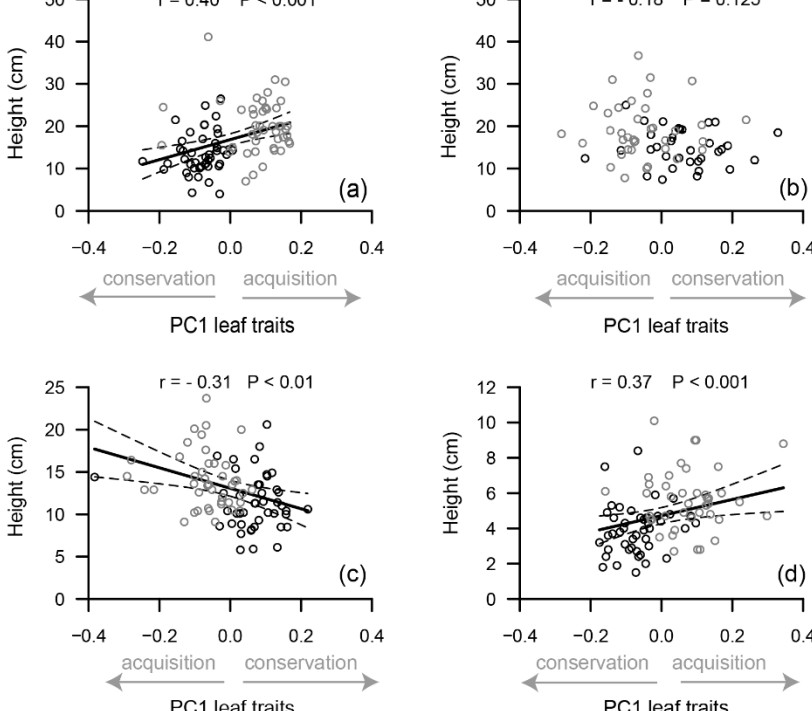




**Figure 5: Relationship between plant height and the first axis of the principal component analysis (PC1) of leaf traits for *Betula nana* (a), *Salix pulchra* (b), *Ledum palustre* (c) and *Vaccinium vitis-idaea* (d). Points are trait values for individuals on unfertilized (black) and fertilized plots (grey). Solid lines are values predicted by the linear mixed-effect model and dashed lines are the upper and lower limits of the predicted value confidence interval. Pearson's correlation coefficient (r) and p-value (P) are indicated on each panel. Main leaf traits comprising PC1 are indicated by grey arrows on the x-axis and grouped into leaf resource acquisition (higher SLA, LNC and LPC) and conservation traits (higher LDMC and leaf C:N).**