# Peer review of "Plant trait response of tundra shrubs to permafrost thaw and nutrient addition"

_Biogeosciences, 2019_

## Short Comment (SC1) · 19 Apr 2020

I found the manuscript clear and well written. I think it provides a nice contribution to climate change impacts in the arctic system. Here are a few line-by-line comments.

L12: I would not use the word adapt as that implies a genetic response, which is not tested here. L62: I find this reference to Voille odd. I would say performance traits (detailed in Voille et al. 2007). L66: Remove 'however' L120: When were these sampled? At the beginning and end of the experiment? Only after the 4 years? L168 & L174: The model is actually block, heat treatment, fert treatment and treatment interaction. Not solely the interaction. I think the description of the analysis could use a bit of revision for clarity. It is often written in a condensed way that makes it tough to

follow. I would add a bit of text to improve clarity of the model descriptions, especially for the contrasts. L203: I am not sure 'no-growing' is correct, maybe 'dormant' L266-267: Shrubs are plants, so this is redundant. L269: Confusing. Try : "Shrubs were released from growth limitation via nutrient addition, which was evidenced by..." or "Nutrient addition released shrubs from growth limitation as evidenced by..." L289-291: Which species would outcompete shrubs in this system? Which species will shade them out? L301: evergreen and deciduous are not species but PFT and if you mean species then use among not between. L310: Did you test wood density in tissues grown before and after treatment? It seems like the sampling strategy would allow partitioning the inner and outer stem to see differences. L320: It is not clear regarding the point that stem and leaf trade-offs operate independently. Please revise for clarity. L349: Remove 'also' L357: Confusing sentence. L322-370: I found the discussion overly speculative in an effort to relate the trait responses into a climate-vegetation feedback. I would encourage the authors reduce the speculation or possibly present the information as potential scenarios of climate and vegetation responses. Fig 5: Could you inverse two of the PC1 values so that the x-axis is always conservative on the left side and acquisitive on the right? It would make it easier to read.

---

## Referee Comment (RC1) · Tariq Munir (Referee) · 25 Apr 2020

General comments This manuscript is very well articulated, and it will attract readers working on plant PFT/traits responses to permafrost thaw and subsequent nutrient concentration in gelisols in the event of climate warming. The paper adds to our knowledge about how shrubs will respond to climate warming with their strategies to fight back by trading-off between traits for their sustained growth. The manuscript does not describe how many times and how frequent the experimental sites were visited during the study years to imagine the field-work extent of this manuscript which tries to provide many solid conclusions. Without this information, it looks like the sites were set up and flowed by a couple of campaigns each year. No pre-experimental conditions of the selected blocks are provided (if they were similar of different with some

statistical analyses) which could be another drawback of this research. The statistical analyses/models performed might need a quick look back or rerun with random effects. There are no repeated measurements over the years I know of.

Specific comments Paras 2-3: resource acquisition and conservative strategies of plants. do authors have references to these strategies studied Line 52. Would you like to put a reference for projected conditions? Line 54. I would better put a semi-colon here instead of two parentheses Line 62. Please correct referencing here I know, one can derive specific objectives form the last paragraph of hypothesis and an overview of the experimental components; however, I would better explicitly mention specific objectives helpful for researchers skimming several studies at a time Line97. Define growing season (e.g., May-Oct); I think, it must be a point when the daily maximum temperature reaches a minimum of 6-degree C. The growing season was never defined except table 1 Line102. What was the extent of a block? A schematic may help here Line105. How heating cables were buried? I hope sunny days were long enough to charge batteries and the batteries never failed Line166-170. I am afraid the random effects of the plot was not included in lme Line 350-368. Discussion tries to relate no matter what I am trying to understand why this experiment could not be completed in less than four years when the year seems not to have any specific function, for example, repeated measurements? I know the fertilizer was applied twice, the second time after two years – other than that do not know why four years are emphasized? Stem trait did not show response even after four years anyway I do not see tables S2 S3 and fig. 5 s1 mentioned in the text

---

## Referee Comment (RC2) · Michael Klinge (Referee) · 26 Jun 2020

General comments: In this work the authors present their results of bio-ecological investigation on tundra shrubs. They used an experimental setup for examining the change of physiological plant conditions induced by actual climate change. The basic assumption was that climate warming leads to enhanced permafrost thaw, which simultaneously will add nutrients to the system. Leaf and stem traits of four different shrubs species were statistically analysed. The manuscript is generally well structured, clearly written and substantially justified by literature. The results about the potential adjustments of plant growing strategies contribute new insights for future environmental development in the subarctic region.

Specific comments: There are two general obstacles in the experiment setup, which I propose to consider more for discussion and conclusion: A main result of the experiment was that no significant response of plant traits was found due to permafrost thaw, whereas significant plant-trait response to fertilization was proofed. The general assumption was that the nutrient supply will increase caused by enhanced mineralization and thawed soil due to climate warming. This means that solely the soil heating would already lead to an increase of nutrient supply during the experiment. I am wondering, why an effect of increased nutrient supply is not observed in the data for the solely heating part of the experiment. Parallel soil analyses would have underlined the presumed causal chains. The fertilization part of the experiment represents an extraordinary nutrient input into the system, which are marginal under natural conditions, when a slight input of nitrified dust from anthropogenic sources, desiccated lakes and desertificated landscapes is taken into account. The heating cables are buried in a depth of 15 cm. This brings along a systematic problem for the study design when compared to expected environmental changes under natural conditions. Climate warming controls soil temperatures along air temperatures. Thus, soil heating begins at the top surface and temperature changes move downward with decreasing amplitude. The high content of organic material in subarctic topsoil has a specific influence on the thermal conductivity into the subsoil. During summer, it may have an isolating effect, when it becomes dried-up; during winter, the thermal conductivity increases caused by soil moisture content.

Technical corrections: L25: Here you are talking of "all four species" but until now you didn't introduce them in the abstract. L59: Please give detailed information about the distinct methods used in "several experiments and satellite imagery" L120: When did you select and cut the individuals? I think after 4 years at the end of the experiment. This should be mentioned here. In addition: Why not selecting the individuals already in the beginning of the experiment; to measure some initial parameters such as plant height and LA? Then you would be able to document relative changes for individuals over the period? L135: Space between 1 and cm2

---

## Author Comment (AC1) · 17 Jul 2020

Referee's comments in black
Authors' responses in blue

**Responses to Michael O'Brien (Short Comment)**

I found the manuscript clear and well written. I think it provides a nice contribution to climate change impacts in the arctic system. Here are a few line-by-line comments.

We thank Michael O'Brien for his time and constructive comments, which will notably improve the clarity of the manuscript.

L12: I would not use the word adapt as that implies a genetic response, which is not tested here.

We will use "change" instead of "adapt" to avoid misunderstandings with genetic responses.

L62: I find this reference to Violle odd. I would say performance traits (detailed in Violle et al. 2007).

OK.

L66: Remove 'however'

OK

L120: When were these sampled? At the beginning and end of the experiment? Only after the 4 years?

The individuals were sampled at the end of the experiment (year 4 of experiment (2014)).We will add a sentence with this information to subsection "2.4 Study species and sampling".

L168 & L174: The model is actually block, heat treatment, fert treatment and treatment interaction. Not solely the interaction. I think the description of the analysis could use a bit of revision for clarity. It is often written in a condensed way that makes it tough to follow. I would add a bit of text to improve clarity of the model descriptions, especially for the contrasts.

We will correct the description of the model and revise it for clarity.

L167-L169: "[…]. The fixed terms of the models were block (factor with five levels), permafrost thaw treatment (two levels: heating, no-heating), fertilization treatment (two levels: fertilization, no-fertilization) and treatment interaction. […]".

L172-L184: " We also used linear mixed-effect models to test the treatment effect on plant traits. Height, LA, bark thickness, and xylem diameter were log-transformed prior the statistical analysis to meet assumptions of linearity. First, we analysed plant traits of the four species together and then plant traits of each functional type (PFT; deciduous and evergreen). In the species analysis, we modelled each plant trait as a function of block (a fixed factor with five levels), permafrost thaw

treatment (a fixed factor with three levels), fertilization treatment (a fixed factor with two levels), species (a fixed factor with four levels) and the interaction between treatments and species. In addition to these fixed terms, we also considered the interaction between species and block, which was a term recognised in the course of the statistical analysis to take into account species-specific trait differences among blocks. The random terms of the model were plot (factor with 30 levels) and the interaction of plot and species. In the PFT analysis, we modelled plant traits as a function of block, permafrost thaw treatment, fertilization treatment, PFT (a fixed factor with two levels), species and the interaction between treatments, PFT and species. The random terms were plot and the interaction between plot and species. In both cases, we assessed if the effects of the permafrost thaw treatment on plant traits were due to the disturbance of the buried cables or the treatment *per se*. For that purpose, we splat the three-level permafrost thaw factor into two contrasts of one degree of freedom (df) each, i.e. cable presence (heated and unheated cables vs. no-cable) and heating (heated cables vs. unheated cable and no-cable). We used the first contrast to assess cable effects (heating followed by cable presence) and the second contrast to assess treatment effects (cable presence followed by heating). After running these models for species and PFTs, we found that plant traits were significantly different among species, even between species within the same PFT. Consequently, we analysed the four species separately to maintain ecological information. We fitted block, permafrost thaw treatment, fertilization treatment and the treatment interaction as fixed terms and plot as a random term".

L203: I am not sure 'no-growing' is correct, maybe 'dormant'

We use the term 'no-growing', which is widely used in Arctic and alpine research. Some examples of this use can be found in Lin *et al.* (2011), Parmentier *et al.* (2011), Rumpf *et al.* (2014), van der Molen *et al.* (2007), Wang *et al.* (2016).

Lin, X., Zhang, Z., Wang, S., Hu, Y., Guangping, X., Luo, C., Chang, X., Duan, J., Lin, Q., Xu, B., Wang, Y., Zhao, X. and Xie, Z. (2011). Response of ecosystem respiration to warming and grazing during the growing seasons in the alpine meadow on the Tibetan plateau. *Agricultural and Forest Meteorology,* 792-802.

Rumpf, S.B., Semenchuck, S.D., Cooper, E.J. (2014). Idiosyncratic responses of High Arctic plants to changing snow regimes. *PLoS ONE*, 9(2): eB6281.

van der Molen, M.K.; van Huissteden, J., Parmentier, F.J.W., Petrescu, A.M.R., Dolman, A.J. *et al.* (2007). The growing season greenhouse gas balance of a continental tundra site in the Indigirka lowlands, NE Siberia. *Biogeosciences*, European Geosciences Union, 4 (6), pp. 985-1003.

Parmentier, F.J.W., van der Mole, M.K., van Huissteden, J., Karsanaev, S.A., Kononov, A.V., Suzdalov, D.A., Maximov, T.C. and Dolman, A.J. (2011). Longer growing seasons do not increase net carbon uptake in the northeastern Siberian tundra. *Journal of Geophysical Research*, 116, G04013.

Wang, P., Mommer, L., van Ruijven, J., Berendse, F., Maximov, T.C. and Heijmans, M.M.P.D. (2015) Seasonal changes and vertical distribution of root standing biomass of graminoids and shrubs at a Siberian tundra site. *Plant Soil*, 407, 55-65.

L266-267: Shrubs are plants, so this is redundant.

We will rephrase the sentence "Plant growth in high-latitude ecosystems is highly nutrient-limited (Billings & Mooney 1968; …)".

L269: Confusing. Try : "Shrubs were released from growth limitation via nutrient addition, which was evidenced by. . ." or "Nutrient addition released shrubs from growth limitation as evidenced by. . ."

We will change the sentence to "Nutrient addition released shrubs from growth limitations as evidenced by the plant trait changes we found, such as …"

L289-291: Which species would outcompete shrubs in this system? Which species will shade them out?

In Arctic tundra, graminoid species - particularly grass species - are expected to shade and outcompete shrubs, as suggested by results of warming and fertilization experiments carried out on tundra sites (Dormann and Woodin 2002, Gough and Hobbie 2003). In the study area, the species that are expected to outcompete shrubs are *Calamagrostis holmii* Lange and *Eriophorum vaginatum* L. (Wang *et al. 2017).* We will add a sentence to the discussion to answer both questions.

Dormann, C.F. and Woodin, S.J. (2002). Climate change in the Arctic: using plant functional types in a meta-analysis of field experiments. *Functional Ecology,* 16, 4−17.

Gough, L. and Hobbie, E. (2003). Responses of moist non-acidic arctic tundra to altered environment: productivity, biomass, and species richness. *Oikos*, 103, 204−216.

Wang, P., Limpens, J., Mommet, L., van Ruijven, J., Nauta, A.L., Berendse, F., Schaepman-Strub, G., Blok, D., Maximov, T.C., Heijmans, M.M.P.D. (2017). Above- and below-ground responses of four tundra plant functional types to deep soil heating and surface soil fertilization. *Journal of Ecology*, 105, 947−957.

L301: evergreen and deciduous are not species but PFT and if you mean species then use among not between.

The sentence will be corrected: "…in the leaf economics spectrum both between PFTs, i.e. from evergreen to deciduous … and within species…".

L310: Did you test wood density in tissues grown before and after treatment? It seems like the sampling strategy would allow partitioning the inner and outer stem to see differences.

We did not test wood density in tissues grown before and after treatment. Because of the sampling protocol applied (individuals were selected at the end of experiment), identifying tissues growing before and after the treatments were difficult. However, in another study based on this experiment (Iturrate-Garcia *et al.* 2017), the treatment effects were tested on inner and outer stem variables (i.e. bark thickness, xylem diameter, bark investment, wood biomass). In addition, annual growth rings of the four years before the experiment and the four years of experiment were tested, resulting in a significant increase on the distance between rings (i.e. growth rate) on shrubs growing on fertilized plots.

Iturrate-Garcia, M., Heijmans, M.M.P.D., Schweingruber, F.H., Maximov, T.C., Niklaus, P.A., Schaepman-Strub, G. (2017). Shrub growth rate and bark responses to soil warming and nutrient addition – A dendroecolological approach in a field experiment. *Dendrochronologia*, 45, 12-22.

L320: It is not clear regarding the point that stem and leaf trade-offs operate independently. Please revise for clarity.

We will rephrase this sentence and the previous one for clarity.

Line 320: "However, our results showed that coordination between stem-height PC1 and leaf PC1 was only significant for half of the species. For *Betula nana* and *Vaccinium vitis-idaea*, the significant relationship between both axes suggested that these species coordinated stem and leaf traits (e.g. conservative trade-off at stem and leaf levels), resulting in a whole-plant strategy. The lack of coordination between stem and leaf traits for *Salix pulchra* and *Ledum pallustre* suggests that, for certain species, functional trade-offs at stem and leaf levels may operate partly independent (Fortunel, Fine & Baraloto, 2012)."

L349: Remove 'also'

OK

L357: Confusing sentence.

We will revise the sentences included in Lines 356-357 for clarity:

L356-357: "This depletion might result in reduced permafrost thaw through decreasing soil moisture, thermal conductivity, heat flux and temperature, which suggests that shrub shading might not be the only driver of the reduced permafrost thaw."

L322-370: I found the discussion overly speculative in an effort to relate the trait responses into a climate-vegetation feedback. I would encourage the authors reduce the speculation or possibly present the information as potential scenarios of climate and vegetation responses.

This part of the discussion is meant to highlight how the results of this study based on detailed trait analyses and plant strategies support earlier findings that only covered part of the leaf and plant economic spectrum. We will revise the language of the indicated discussion section to address the reviewer's comment.

Fig 5: Could you inverse two of the PC1 values so that the x-axis is always conservative on the left side and acquisitive on the right? It would make it easier to read.

We will change Figure 5 to have the conservative strategy on the left side and acquisitive on the right side.

---

## Author Comment (AC2) · 17 Jul 2020

Referee's comments in black
Authors' responses in blue

**Responses to Tariq Munir (Referee 1)**

**General comments**

This manuscript is very well articulated, and it will attract readers working on plant PFT/traits responses to permafrost thaw and subsequent nutrient concentration in gelisols in the event of climate warming. The paper adds to our knowledge about how shrubs will respond to climate warming with their strategies to fight back by trading-off between traits for their sustained growth.

We thank Tariq Munir for taking the time to read and comment on our manuscript.

The manuscript does not describe how many times and how frequent the experimental sites were visited during the study years to imagine the field-work extent of this manuscript which tries to provide many solid conclusions. Without this information, it looks like the sites were set up and flowed by a couple of campaigns each year.

The experimental site was visited periodically during the study years as described in the reference we included in Line 101 (Wang *et al.*, 2017). The campaigns were limited to the growing season; due to the harsh conditions of the study area, the experimental plots were not accessible during the rest of the year. Below, we enumerate the visits to the plots (information extracted from Wang *et al.*, 2017).

The experimental plots were selected in July 2010; the heating cables were inserted in the soil also in this period. The following growing season (July 2011), the experimental treatments (heating and fertilization) were implemented in July 2011: the heating cables were connected to the solar panels in the heated plots and slow-release NPK fertilizer tablets were added to the fertilized plots. In 2013, nutrients were added again to the plots. During the experimental period, environmental factors were measured periodically. The permafrost thaw depth and soil moisture were measured 2-4 times per growing season, while soil temperature was measured continuously (data were recovered from the temperature loggers each growing season). Resin bags to assess soil nutrient availability were inserted in each plot in 2010 and replaced by new ones at the beginning of each August until 2014. The species abundance within the plots was recorded in 2010 and 2013 (results published in Wang et al., 2017). Aboveground and belowground biomass was harvested in August 2014 (Wang *et al.*, 2017). The individuals selected for the plant trait analysis and growth rate (Iturrate-Garcia *et al.*, 2017) were sampled in the second half of 2014 growing season (31 July - 12 August 2014), when leaf and stem traits were also measured (indicated in Lines 128-129).

The above information will be added to section 2 Materials and Methods (subsections 2.2 Experimental design and 2.4 Study species and sampling).

Iturrate-Garcia, M., Heijmans, M.M.P.D., Schweingruber, F.H., Maximov, T.C., Niklaus, P.A., Schaepman-Strub, G. (2017). Shrub growth rate and bark responses to soil warming and nutrient addition – A dendroecolological approach in a field experiment. *Dendrochronologia*, 45, 12-22.

Wang, P., Limpens, J., Mommer, I., van Ruijven, J., Nauta, A.L., Berendse, F., Schaepman-Strub, G., Blok, D., Maximov, T.C., Heijmans, M.M.P.D. (2017). Above and belowground responses of four tundra plant functional types to deep soil heating and surface soil fertilization. *Journal of Ecology*, 105, 164−175.

No pre-experimental conditions of the selected blocks are provided (if they were similar of different with some statistical analyses) which could be another drawback of this research.

Although no pre-experimental condition of the selected blocks are provided, potential differences within blocks were statistically considered by including "block" (factor with 5 levels) as fixed term in all the linear mixed-effect models. Moreover, in order to take into account species-specific trait differences among blocks, we added the interaction between species and block (fix terms) to the statistical models, which we run for plant traits (detailed in subsection "2.6 Statistical analysis").

The statistical analyses/models performed might need a quick look back or rerun with random effects. There are no repeated measurements over the years I know of.

Except for the statistical models used to test the effects of treatments on soil temperature and permafrost thaw depth (see explanation at specific comment further down, referring to Line 166-170), the other statistical models (effects of treatments on plant traits) consider plot as random effect, as described in subsection "2.6 Statistical analysis". It is correct that there are no repeated measurements over the years.

**Specific comments**

Paras 2-3: resource acquisition and conservative strategies of plants. do authors have references to these strategies studied

We will add the reference of Diaz *et al.* (2016) on Line 44, which refers to acquisitive and conservative strategies of plants.

Díaz, S., Kattge, J., Cornelissen, J.H.C., Wright, I.J., Lavorel, S., Dray, S. et al. (2016). The global spectrum of plant form and function, *Nature*, 529, 1-17.

Line 52. Would you like to put a reference for projected conditions?

We will add the article by Post *et al.* (2019) on Line 52, as a reference for projected conditions in the Arctic.

Post, E., Alley, R.B., Christensen, T.R., Macias-Fauria, M., Forbes, B.C., Gooseff, M.N., Iler, A., Kerby, J.T., Laidre, K.L., Mann, M.E., Olofsson, J., Stroeve, J.C., Ulmer, F., Virginia, R.A., Wang, M. (2019). The polar regions in a 2°C warmer world. *Science Advances,* 5, eaaw9883.

Line 54. I would better put a semi-colon here instead of two parentheses

OK

Line 62. Please correct referencing here I know, one can derive specific objectives form the last paragraph of hypothesis and an overview of the experimental components; however, I would better explicitly mention specific objectives helpful for researchers skimming several studies at a time

If we understood correctly, in this comment Taquir Munir (Referee 1) raises his concern about using solely the reference Violle et al. (2007) instead of adding several studies at a time. The inclusion of this reference, however, aims to clarify the concept of trait used in the studies related to tundra shrub responses to climate change. Violle et al. (2007) is a theoretical/review paper. This paper introduces the concept of "performance traits" and specifically elaborates on traits included in this category. Addressing the suggestion of the received Short Comment by Michael O'Brien (L62), we rephrased the sentence ('[…] performance traits (detailed in Violle et al., 2007) […]"), which might help to address also the comment by Taquir Munir.

In case that this comment refers to Line 74 instead of Line 62, we will rephrase the sentence to specifically mention the objectives of the study. 'The objective of this study is to experimentally investigate…'

Line97. Define growing season (e.g., May-Oct); I think, it must be a point when the daily maximum temperature reaches a minimum of 6-degree C. The growing season was never defined except table 1

We use the term "growing season" for the period of the year during which arctic plants photosynthesize and grow. The timing of the growing season depends on the air temperature, snowmelt, and thawing of frozen soil at the study site. Air temperature (e.g. 6°C) alone is therefore not an appropriate proxy for the growing season and it can vary from year to year (± 4 days; Parmentier *et al.* 2011) and from species to species. For the research area at the ecosystem level, the growing season lasts approximately from end of June to end of August, based on carbon flux analyses (Parmentier *et al.* 2011, van der Molen *et al.* 2007).

To address the referee's comment we will add an indication of time on Line94: '...and lichens. The growing season lasts from the end of June to end of August. The slightly acidic soil...'

Parmentier, F.J.W., van der Mole, M.K., van Huissteden, J., Karsanaev, S.A., Kononov, A.V., Suzdalov, D.A., Maximov, T.C., Dolman, A.J. (2011). Longer growing seasons do not increase net carbon uptake in the northeastern Siberian tundra. *Journal of Geophysical Research*, 116, G04013.

van der Molen, M.K.; van Huissteden, J., Parmentier, F.J.W., Petrescu, A.M.R., Dolman, A.J. *et al.* (2007). The growing season greenhouse gas balance of a continental tundra site in the Indigirka lowlands, NE Siberia. *Biogeosciences*, European Geosciences Union, 4 (6), pp. 985−1003.

Line102. What was the extent of a block? A schematic may help here

The six plots within each block were spaced by 1-2 meter distance from each other. With a size of 1.5 m x 1.5 m, all plots within a block were located in an area of approximately 10 m x 10 m. A picture of a typical block setup is contained in the supplementary material of Wang *et al.,* 2017. We will add the approximate extent of blocks to the experimental design section.

Wang, P., Limpens, J., Mommer, I., van Ruijven, J., Nauta, A.L., Berendse, F., Schaepman-Strub, G., Blok, D., Maximov, T.C., Heijmans, M.M.P.D. (2017). Above and belowground responses of four tundra plant functional types to deep soil heating and surface soil fertilization. *Journal of Ecology*, 105, 164−175.

Line105. How heating cables were buried? I hope sunny days were long enough to charge batteries and the batteries never failed

A detailed description of the experimental set-up and burying of the heating cables, including pictures, is provided in the reference indicated in L101 (Wang *et al.* 2017, including

supplementary materials). The heating cables were buried into the soil from trenches excavated at two opposing sides of the experimental plots to minimize disturbance within the plots (i.e. disturbance of roots and microbial activity). For the heated cable treatment, the cables were connected to two solar panels of 85 W each, which were connected in parallel. The solar panels were installed with an angle of 60° to capture 20 hours of sunlight per day during the summer. No battery was included in the circuit. Thereby, the solar energy directly enlarged the natural ground heat flux, allowing for diurnal and seasonal variation in solar intensity.

Wang, P., Limpens, J., Mommer, I., van Ruijven, J., Nauta, A.L., Berendse, F., Schaepman-Strub, G., Blok, D., Maximov, T.C., Heijmans, M.M.P.D. (2017). Above and belowground responses of four tundra plant functional types to deep soil heating and surface soil fertilization. *Journal of Ecology*, 105, 164−175.

Line 166-170. I am afraid the random effects of the plot was not included in lme

For this statistical analysis − effects of permafrost thaw and fertilization treatments on soil temperature and thaw depth of the experimental plots – we used the average of the permafrost thaw depth and temperature data per plot as response variable. As indicated in subsection 2.3 of the manuscript, data were collected in 2013 (only temperature) and 2014 (temperature and thaw depth). Because we aggregated the data per experimental plot, we removed the random term (plot) from our analysis, in order to avoid model-overfitting.

Line 350-368. Discussion tries to relate no matter what I am trying to understand why this experiment could not be completed in less than four years when the year seems not to have any specific function, for example, repeated measurements? I know the fertilizer was applied twice, the second time after two years – other than that do not know why four years are emphasized? Stem traits did not show response even after four years anyway.

Although repeated measurements were not used in this study, the experimental duration over several years (i.e. 4 years) was important for many reasons. By running the experiment for four years, we aimed at reducing potential disturbance effects by the initial experimental setup (i.e. soil disturbance due to introduction of heating cables). Moreover, we selected a duration of several years considering the characteristics of tundra plant species, i.e. low rates of resource acquisition, growth and tissue turnover (Chapin 1980). This consideration was especially important for the shrub growth rate study associated with the experiment (Iturrate-Garcia et al., 2017). In this case, the experimental duration covered different climate conditions in the control plots across years and multiple annual growth rings. This fact allowed us to get insights into the effects of climatic conditions on growth rate (unpublished results).

As indicated in the answer addressing Referee 2's comment (L120), the aim of this study is to test treatment effects on plant traits by comparing treatment plots with control plots after 4 years of experiment and not to document relative changes for individuals over the experimental period. Because of this aim and the destructive sampling of individuals needed for the current study, we did not have repeated measurements across the years. Furthermore, we sampled only at the end of the experiment to avoid disturbing the setup.

The lack of response of stem traits, indeed, might be explained by the relative short term of the experiment, as explained in Lines 307-309. Wood tissue turnover is slower than leaf tissue (Negrón-Juárez et al., 2015). Thus, stem traits might require even more time (> 4 years) to show responses.

Chapin, F.S. III (1980). The mineral nutrition of wild plants. *Annual Review of Ecology and Systematics*, 11, 233−260.

Iturrate-Garcia, M., Heijmans, M.M.P.D., Schweingruber, F.H., Maximov, T.C., Niklaus, P.A., Schaepman-Strub, G. (2017). Shrub growth rate and bark responses to soil warming and nutrient addition − A dendroecological approach in a field experiment. *Dendrochronologia*, 45, 12−22.

Negrón-Juárez, R.I., Koven, C.D., Riley, W.J., Knox, R.G., Chambers, J.Q. (2015). Observed allocations of productivity and biomass, and turnover times in tropical forests are not accurately represented in CMIP5 Earth system models. *Environmental Research Letters,* 10, 064017.

I do not see tables S2 S3 and fig. 5 s1 mentioned in the text

The tables S2 –S3 and figure S1 mentioned in the text are included in the supplementary material: https://www.biogeosciences-discuss.net/bg-2019-498/bg-2019-498-supplement.pdf

---

## Author Comment (AC3) · 17 Jul 2020

Referee's comments in black
Authors' responses in blue

**Responses to Michael Klinge (Referee 2)**

**General comments**

In this work the authors present their results of bio-ecological investigation on tundra shrubs. They used an experimental setup for examining the change of physiological plant conditions induced by actual climate change. The basic assumption was that climate warming leads to enhanced permafrost thaw, which simultaneously will add nutrients to the system. Leaf and stem traits of four different shrubs species were statistically analysed. The manuscript is generally well structured, clearly written and substantially justified by literature. The results about the potential adjustments of plant growing strategies contribute new insights for future environmental development in the subarctic region.

We thank Michael Klinge for his time to review our manuscript and for providing constructive and relevant comments, which will improve our manuscript.

**Specific comments**

There are two general obstacles in the experiment setup, which I propose to consider more for discussion and conclusion: A main result of the experiment was that no significant response of plant traits was found due to permafrost thaw, whereas significant plant-trait response to fertilization was proofed. The general assumption was that the nutrient supply will increase caused by enhanced mineralization and thawed soil due to climate warming. This means that solely the soil heating would already lead to an increase of nutrient supply during the experiment. I am wondering, why an effect of increased nutrient supply is not observed in the data for the solely heating part of the experiment. Parallel soil analyses would have underlined the presumed causal chains.

We agree on the expected increase of nutrient supply in heating plots through two potential mechanisms, namely enhanced mineralization (if soil temperature increases) and thawed soil material containing nutrients. Parallel soil analyses have been performed and are reported in Wang *et al.*, 2017 (supplementary material). These analyses based on buried resin bags showed no increase of exchangeable nutrients in the non-fertilized unheated and heated plots for nitrogen and phosphorus. In the fertilized plots an increase in the top soil (5cm) was found for nitrogen (4x) and phosphorus (5x), and only slightly (but significantly) increased at a depth of 25 cm for nitrogen, but not for phosphorus. Increasing the energy input into the soil might result in a very low increase of temperature under humid conditions (i.e. high soil thermal conductivity) close to the permafrost table, where the cables were buried, as the energy might be partitioned towards the thawing process, and not towards heating the soil. Hence, the increase in mineralization rate might be rather limited in the plots with heated cables due to limited increase in temperature.

We will add a corresponding short section to the discussion, as suggested by the reviewer.

Wang, P., Limpens, J., Mommer, I., van Ruijven, J., Nauta, A.L., Berendse, F., Schaepman-Strub, G., Blok, D., Maximov, T.C., Heijmans, M.M.P.D. (2017). Above and belowground responses of four tundra plant functional types to deep soil heating and surface soil fertilization. *Journal of Ecology*, 105, 164−175.

The fertilization part of the experiment represents an extraordinary nutrient input into the system, which are marginal under natural conditions, when a slight input of nitrified dust from anthropogenic sources, desiccated lakes and desertificated landscapes is taken into account. The heating cables are buried in a depth of 15 cm. This brings along a systematic problem for the study design when compared to expected environmental changes under natural conditions. Climate warming controls soil temperatures along air temperatures. Thus, soil heating begins at the top surface and temperature changes move downward with decreasing amplitude. The high content of organic material in subarctic topsoil has a specific influence on the thermal conductivity into the subsoil. During summer, it may have an isolating effect, when it becomes dried-up; during winter, the thermal conductivity increases caused by soil moisture content.

The fertilization indeed represents a large nutrient input into the ecosystem, much larger than the atmospheric nutrient input. The goal of adding fertilizer was to mimic future increased nutrient availability in the soil resulting from increased mineralization of nutrients in the soil organic layer in future warmer soils. The nutrient input is perhaps extraordinary compared to the atmospheric input, which is extremely low in the Arctic, but less so compared to soil nutrient availability and other fertilization experiments in Arctic tundra.

The fertilizer was added in the form of slow-release tablets. As the release rate depends on soil temperature, which is low at our study site, we applied a rather high dose. This dose should be seen as the maximum, probably not all of the nutrients have been released. When we performed measurements in the plots two years after application, we found some intact tablets in dry moss, indicating not all of the nutrients added had become available.

As stated by the reviewer, soil moisture conditions along the vertical profile will impact the energy partitioning within the soil towards heating and permafrost thawing. As mentioned above, the aim of this experiment was to simulate permafrost thaw and not soil warming. As stated in Wang *et al.,* 2017, 'The deep soil heating treatment increased June–July thawing depth without increasing soil temperatures in the upper organic soil layer, for the first time enabling us to separate the effects of increased thawing depth from the effects of surface soil environmental changes in the tundra.'

We will add a section in the introduction to make this more explicit.

Wang, P., Limpens, J., Mommer, I., van Ruijven, J., Nauta, A.L., Berendse, F., Schaepman-Strub, G., Blok, D., Maximov, T.C., Heijmans, M.M.P.D. (2017). Above and belowground responses of four tundra plant functional types to deep soil heating and surface soil fertilization. *Journal of Ecology*, 105, 164−175.

**Technical corrections**

L25: Here you are talking of "all four species" but until now you didn't introduce them in the abstract.

We will add this information to L21: "…in four shrub species (*Betula nana, Salix pulchra, Ledum palustre, Vaccinium vitis-idaea),* which were sampled in the experimental plots".

L59: Please give detailed information about the distinct methods used in "several experiments and satellite imagery"

We will add more information about the distinct methods used in "several experiments and satellite imagery".

"Several warming experiments (Elmendorf *et al.*, 2012), satellite imagery − i.e. AVHRR, MODIS and Landsat multi-decadal records of the normalized difference vegetation index (NDVI) (Myers-Smith et al., 2011) − and repeat multi-decadal aerial photography (Tape *et al.*, 2012) have shown effects of recent climate warming on tundra vegetation growth, productivity and distribution, especially on shrubs (Myers-Smith *et al.,* 2015; Myers-Smith & Hik, 2018)".

Elmendorf, S.C., Henry, G.H.R., Hollister, R.D., Björk, R.G., Bjorkman, A.D., Callaghan, T.V., …, Wookey, P.A. (2012). Global assessment of experimental climate warming on tundra vegetation: heterogeneity over space and time. *Ecology Letters*, 15, 164−175.

Myers-Smith, I.H., Elmendorf, S.C., Beck, P.S.A., Wilmking, M., Hallinger, M., Blok, D., …, Vellend, M. (2015). Climate sensitivity of shrub growth across the tundra biome. *Nature Climate Change*, 5, 887−891.

Myers-Smith, I.H., Forbes, B.C., Wilmking, M., Hallinger, M., Lantz, T., Blok, D., …, Hik, D.S. (2011). Shrub expansion in tundra ecosystems: dynamics, impacts and research priorities. *Environmental Research Letters*, 6, 045509 (15 pp).

Myers-Smith, I.H. and Hik, D.S. (2018). Climate warming as a driver of tundra shrubline advance. *Journal of Ecology*, 106, 887−891.

Tape, K.D., Hallinger, M., Welker, J.M., Ruess, R.W. (2012). Landscape heterogeneity of shrub expansion in Arctic Alaska, *Ecosystems,* 15, 711−724.

L120: When did you select and cut the individuals? I think after 4 years at the end of the experiment. This should be mentioned here. In addition: Why not selecting the individuals already in the beginning of the experiment; to measure some initial parameters such as plant height and LA? Then you would be able to document relative changes for individuals over the period?

Yes, the individuals were selected and sampled at the end of the experiment (year 4 of the experiment (2014)). We will add a sentence with this information to subsection "2.4 Study species and sampling".

The aim of this study was to test treatment effects on plant traits by comparing treatment plots with control plots after 4 years of experiment and not to document relative changes for individuals over the experimental period. Because of this reason, we did not select individuals nor measured individual traits, such as height and LA, at the beginning of the experiment. Furthermore, we focused on assessing treatment effects on groups of traits, instead of on individual traits, to identify changes in shrub strategies. Changes on the selected strategies might provide insight on shrub community shifts in Arctic tundra under future climatic conditions expected for the Arctic, such as rising temperature and increasing permafrost thaw (IPCC 2013, Turetsky *et al.* 2020, Voigt *et al., 2017*).

IPCC: *Climate Change 2013: The Physical Science Basis. Contribution of Working Group I to the Fifth Assessment Report of the Intergovernmental Panel on Climate Change* (eds. TF Stocker, D Qin, G-K Plattner, M Tignor, SK Allen, J Boschung, A Nauels, Y Xia, V Bex, and PM Midgley), Cambridge, United Kingdom and New York, NY, USA.

Turetsky, M.R., Abbot, B.W., Jones, M.C., Anthony, K.W., Olefeldt, D., Schuur, E.A.G., Grosse, G., Kuhry, P., Hugelius, G., Koven, C., Lawrence, D.M., Gibson, C., Sannel, A.B.K., McGuire, A.D. (2020). Carbon release through abrupt permafrost thaw. *Nature Geoscience*, 13, 138 −143.

Voigt, C., Marushchak, M.E., Lamprecht, R.E., Jackowicz-Korcyński, Lingren, A., Mastepanov, M., Granlund, L., Cristensen, T.R., Tahvanainen, T., Martikainen, P.J., Biasi, C. (2017). Increased nitrous oxide from Arctic peatlands after permafrost thaw. *PNAS*, 114 (24), 6238−6243.

L135: Space between 1 and cm2

OK.